# BMJ Open  Long-term trajectories of back pain: cohort study with 7-year follow-up

Kate M Dunn, Paul Campbell, Kelvin P Jordan

## ABSTRACT

**Objective:** To describe long-term trajectories of back pain.

**Design:** Monthly data collection for 6 months at 7-year follow-up of participants in a prospective cohort study.

**Setting:** Primary care practices in Staffordshire, UK.

**Participants:** 228 people consulting their general practitioners with back pain, on whom information on 6-month back pain trajectories had been collected during 2001–2003, and who had valid consent and contact details in 2009–2010, were contacted. 155 participants (68% of those contacted) responded and provided sufficient data for primary analyses.

**Outcome measures:** Trajectories based on patients' self-reports of back pain were identified using longitudinal latent class analysis. Trajectories were characterised using information on disability, psychological status and presence of other symptoms.

**Results:** Four clusters with different back pain trajectories at follow-up were identified: (1) no or occasional pain, (2) persistent mild pain, (3) fluctuating pain and (4) persistent severe pain. Trajectory clusters differed significantly from each other in terms of disability, psychological status and other symptoms. Most participants remained in a similar trajectory as 7 years previously (weighted κ 0.54; 95% CI 0.42 to 0.65).

**Conclusions:** Most people with back pain appear to follow a particular pain trajectory over long time periods, and do not have frequently recurring or widely fluctuating patterns. The results are limited by lack of information about the time between data collection periods and by loss to follow-up. However, findings do raise questions about standard divisions into acute and chronic back pain. A new framework for understanding the course of back pain is proposed.

Arthritis Research UK Primary Care Centre, Institute of Primary Care and Health Sciences, Keele University, Newcastle, Staffordshire, UK

**Correspondence to**
Dr Kate M Dunn;
k.m.dunn@keele.ac.uk

### Strengths and limitations of this study

- The study benefits from long-term follow-up, prospective design, frequent follow-up during study periods, robust analyses and use of validated questionnaire instruments.
- The study was limited by loss to follow-up, meaning restricted number of participants for full analysis, but multiple imputation was used to investigate the implications of this.
- Data collection phases were 7 years apart, and similar information about trajectories in the interim period is unavailable.

indicating that even if not constantly present, back pain is a long-term experience. This has led to a suggestion to use a longer term, life course approach to study back pain.[4]

The long-term experience of back pain is often not addressed by researchers. In a recent review of back pain prognosis, only 1 of the 33 included studies had follow-up beyond a year.[5] Studies with shorter term follow-up can only represent a compressed view of the long-term pain experience. The few longer term studies have limited number of follow-up points.[6–8] Knowledge of prognosis is important, as stratifying back pain management based on risk of poor prognosis can be clinical and cost-effective,[9] with benefits for targeting early treatment and referrals. However, previous research is unable to fully reflect the detailed course of back pain over time or inform about long-term prognosis.

In 2001–2003, we studied a cohort of people consulting in primary care with back pain.[10] We identified four distinct clusters of people with different trajectories: (1) recovering, (2) persistent mild, (3) fluctuating pain and (4) severe chronic back pain. Duration of back pain at baseline increased with rising severity of trajectory, potentially indicating phases of increasing severity in the long-term course. This is supported by models of stages of back pain chronicity[11] and degeneration with age.[12] Alternatively, trajectories could represent distinct groups with stable long-term pain. We aimed to describe long-term trajectories of

## INTRODUCTION

Back pain is common—it has been recently highlighted as the single leading cause of years lived with disability worldwide[1] and many people experience pain over long periods. Among primary care consulters, 38% report having their symptoms for over 3 years.[2] Even among people in primary care with acute back pain, 75% report previous back pain,[3]

back pain through a 6-month follow-up period of a cohort of patients studied 7 years earlier.

## METHODS

This is a follow-up of participants in a back pain cohort study whose short-term (6-month) back pain trajectories had been derived in 2001–2003.[10]

### Study participants

The original study identified people aged 30–59 years consulting with back pain at one of five general practitioners in North Staffordshire, UK, during 2001–2002. Full details are published elsewhere.[13] Briefly, participants returning baseline questionnaires and consenting to follow-up were sent monthly questionnaires. Those returning four or more questionnaires during the first 6 months were included in a longitudinal latent class analysis to determine trajectories of back pain.[10] Of the 342 participants in this original analysis, 73% (n=250) gave their consent to be contacted again. In 2009, current contact details were not available for 22 (6%), leaving 228 people from the original analysis invited to take part at 7-year follow-up.

### Data collection at 7 years

Self-completion questionnaires were mailed to the 228 study participants (7-year baseline mailing) with reminders at 2 and 4 weeks, and brief questionnaires for non-responders at 6 weeks. Participants giving informed consent were sent brief monthly questionnaires for 6 months (the same data collection technique as the original study).

All questionnaires contained the same key measures. Pain intensity was measured using the mean of three 0–10 numerical rating scales.[14] Disability was measured using the modified 23-item Roland-Morris Disability Questionnaire (RMDQ).[15] These instruments were used in the original study,[10] and there is evidence of reliability in the UK primary care back pain patients.[16] The Chronic Pain Grade classified individuals into grades of chronic pain[17]; this was included in the brief 7-year baseline mailing for non-responders. Back pain duration was recalled time since the last pain-free month.[18]

The Hospital Anxiety and Depression Scale (HADS) was used to assess psychological status.[19] It produces scores from 0 to 21, with higher scores indicating more severe symptoms. Insomnia was defined as reporting having trouble falling or staying asleep, waking up several times at night or waking up feeling tired on most nights.[20] This definition has been used previously in pain samples.[21] Somatic symptoms were measured using the 15-item Patient Health Questionnaire (PHQ-15)[22] which is scored from 0 (not bothered with any symptoms) to 30 (bothered a lot with 15 symptoms). Leg pain was self-reported pain travelling from the back to the leg(s), and upper body pain was self-reported pain in the shoulder, arm, neck or head, during the previous 2 weeks.

### Analysis

Two primary analysis groups were formed from responders to this 7-year follow-up study. Group 1 participants returned the 7-year baseline questionnaire plus three or more questionnaires from months 1 to 6. Group 2 included participants with insufficient 7-year follow-up data for full analyses, but who provided adequate information for multiple imputation to be carried out.

For group 1 participants, monthly back pain intensity scores were trichotomised into no pain (scoring less than 1), mild-moderate pain and high pain (scored 5 or more). Longitudinal latent class analysis was used to group participants into clusters based on the trajectory of their back pain over these 6 months as in the original study.[10] In the longitudinal latent class analysis, each participant was allocated to the cluster best matching their pain profile, based on each participant's probability of belonging to each cluster, with participants allocated to the cluster for which they have the largest probability. Participants should be clearly assigned to a single cluster with high probability. Cluster-specific probabilities of having each level of pain for each month, given membership of that cluster, allow development of pain pathways for each cluster. See appendix for more details.

For group 2 participants, the multiple imputation procedure in Stata/IC V.11.1 software with 50 imputations, through a multinomial logistic regression, was used to impute membership of the 7-year clusters identified for group 1. Information on clusters from the original study and outcome measures from the 7-year baseline questionnaire were used to impute cluster membership.

Membership of clusters from both study phases (original and 7-year follow-up) was compared to investigate long-term patterns of trajectory membership. Stability of cluster membership was assessed using weighted κ. The κ can be interpreted as agreement (stability) between original and 7-year follow-up cluster memberships beyond chance, with values of 1 indicating perfect agreement and 0 indicating agreement no better than chance. The 7-year derived clusters (actual or imputed) were compared on the key measures of the 7-year baseline questionnaire, using simple linear or logistic regression as appropriate through the multiple imputation estimate commands in Stata/IC V.11.1.

In order to address potential issues from loss to follow-up from the original 2001–2003 trajectories analysis, an additional group 3 was formed. This included everyone from the original analysis who was not included in the primary analysis at 7 years (above): 7-year responders who provided insufficient data, non-responders at 7 years, people who could not be traced and those not giving consent to follow-up. Groups 1 and 2 combined were compared with group 3 on baseline demographic, pain, anxiety and depression from the original study using t tests or $\chi^2$ tests as appropriate.

As sensitivity analysis, 7 year cluster membership was imputed for groups 2 and 3 participants using information from the original study (baseline RMDQ, Chronic Pain Grade, pain duration and original longitudinal latent class analysis cluster). Comparisons between the original cluster and 7 year actual or imputed cluster membership for participants across all three groups were performed.

## RESULTS

Primary analyses were carried out on 155 responders (68% of the 228 contacted): 112 in group 1 (full data available) and 43 in group 2 (imputation required).

### Clusters at 7-year follow-up

The optimal number of clusters resulting from longitudinal latent class analysis was four (see appendix). Eighty-four per cent of group 1 participants had an average probability of greater than 0.90 of being allocated to their assigned cluster, indicating distinct classification. Group 2 participants were allocated to these clusters using multiple imputation.

The estimated probability of monthly levels of pain within clusters is shown in table 1. These monthly probabilities of pain can be interpreted to describe the occurrence of pain, for example, a probability of mild-moderate pain of 0.13 at baseline for the first cluster indicates that one in every eight people in that group are likely to have experienced mild–moderate pain that month. The first cluster identified (31% of groups 1 and 2) mostly had no pain (estimated monthly probabilities of no pain 0.65–0.87), with occasional mild episodes (cluster labelled 'no or occasional pain'). Participants in this cluster generally reported no pain on at least four occasions over 6 months and did not report high pain. The second cluster (37%) had mild pain intensity most of the time, with a maximum of 1–2 months of no pain; only 17% of the cluster ever reported high pain. Their monthly probabilities of mild pain were between 0.69 and 0.91 ('persistent mild pain'). The third cluster (11%) had pain fluctuating between mild and high levels ('fluctuating pain') and rarely reported no pain. The final cluster (21%) had high-pain intensity levels throughout, with monthly probabilities of high pain between 0.79 and 0.98 ('persistent severe pain') and never reported no pain.

### Comparison of clusters from the original study and 7-year follow-up

The identified trajectories of back pain intensity for the original study and the 7-year follow-up are illustrated in figure 1.

**Table 1** Monthly probability of experiencing each level of back pain based on cluster membership at 7 years

| | Cluster (trajectory) from 7-year follow-up analysis | | | |
| | No/occasional pain | Persistent mild pain | Fluctuating pain | Persistent severe pain |
|---|---|---|---|---|
| Baseline | | | | |
| No pain | 0.87 | 0.15 | 0.01 | 0.00 |
| Mild–moderate pain | 0.13 | 0.80 | 0.51 | 0.21 |
| High pain | 0.00 | 0.05 | 0.48 | 0.79 |
| Month 1 | | | | |
| No pain | 0.85 | 0.06 | 0.00 | 0.00 |
| Mild–moderate pain | 0.15 | 0.91 | 0.62 | 0.17 |
| High pain | 0.00 | 0.04 | 0.38 | 0.83 |
| Month 2 | | | | |
| No pain | 0.65 | 0.06 | 0.00 | 0.00 |
| Mild–moderate pain | 0.35 | 0.89 | 0.12 | 0.11 |
| High pain | 0.00 | 0.05 | 0.88 | 0.89 |
| Month 3 | | | | |
| No pain | 0.70 | 0.07 | 0.01 | 0.00 |
| Mild–moderate pain | 0.30 | 0.86 | 0.58 | 0.17 |
| High pain | 0.00 | 0.07 | 0.42 | 0.83 |
| Month 4 | | | | |
| No pain | 0.66 | 0.09 | 0.00 | 0.00 |
| Mild–moderate pain | 0.34 | 0.88 | 0.29 | 0.18 |
| High pain | 0.00 | 0.03 | 0.71 | 0.82 |
| Month 5 | | | | |
| No pain | 0.75 | 0.26 | 0.19 | 0.00 |
| Mild–moderate pain | 0.25 | 0.69 | 0.73 | 0.02 |
| High pain | 0.00 | 0.05 | 0.09 | 0.98 |
| Month 6 | | | | |
| No pain | 0.80 | 0.16 | 0.02 | 0.00 |
| Mild–moderate pain | 0.20 | 0.79 | 0.66 | 0.11 |
| High pain | 0.00 | 0.05 | 0.32 | 0.89 |

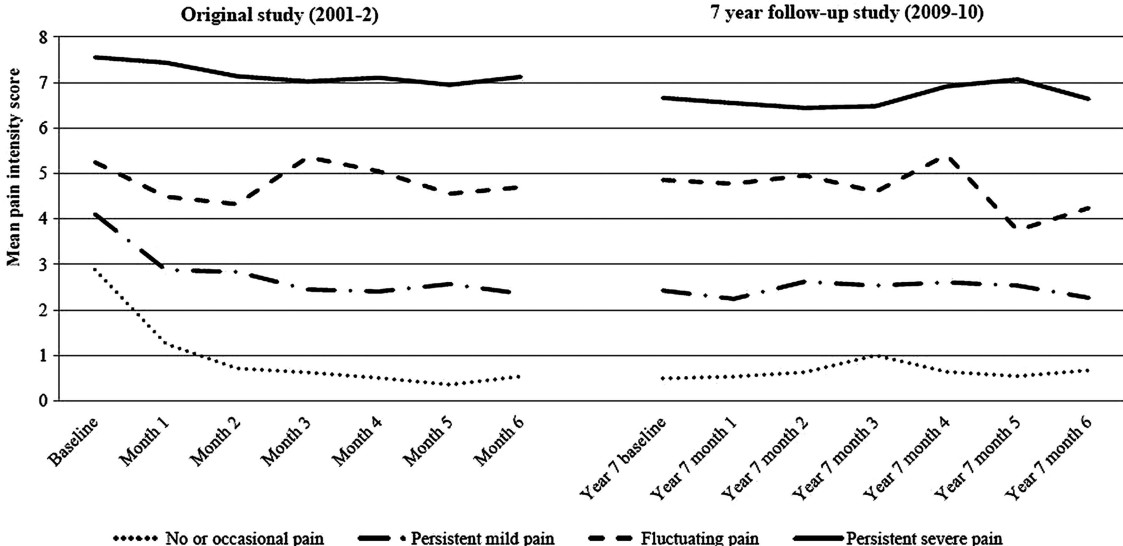

**Figure 1** Trajectories of back pain intensity from original study and 7-year follow-up.

Most of the participants stayed in a similar cluster between the two study phases (weighted κ 0.54 (95% CI 0.42 to 0.65); table 2); 74% (95% CI 57% to 92%) of those originally in the most severe trajectory remained in an equivalent cluster at 7 years. Over half the participants in the two mildest clusters in the original study (recovering: 59%; 95% CI 44% to 74%; persistent mild pain: 56%; 95% CI 40% to 73%) stayed in the most comparable trajectory at 7 years, and most who changed moved to the other mild trajectory. The fluctuating group in the original study (the smallest group) did not show a stable pattern, with 87% of participants changing cluster, mainly to persistent mild or persistent severe clusters.

Pain intensity, disability and psychological status all differed significantly between the 7-year trajectories, with the no or occasional pain cluster having the lowest disability levels (mean RMDQ score 2.0), least pain intensity (mean 0.8) and best psychological status (mean HADS depression score 2.8) and the persistent severe pain cluster having the highest disability (mean RMDQ score 12.9), worst pain intensity (mean 6.7) and poorest psychological status (mean HADS depression score 7.4; table 3). Similar statistically significant differences were also present in the original study.[10] The clusters also

differed significantly in terms of the presence of somatic symptoms and insomnia, with the mean symptom score (PHQ-15) ranging from 3.9 in the no or occasional pain group to 7.7 in the persistent severe pain cluster, and the proportion classified with insomnia ranging from 27% to 80%.

### Sensitivity analyses

Group 3 comprised 25 7-year responders who provided insufficient data, 48 non-responders at 7 years, plus the people from the original study who did not give consent to follow-up (n=92) or could not be traced (n=22). The original study baseline characteristics of the three groups are shown in table 4. The only significant difference between participants in groups 1 and 2 and those in group 3 was gender, with a fewer females in group 3 (p=0.04).

Including imputed data from group 3 as well as group 2 participants made little difference to the estimated relative sizes of the 7-year clusters reported above, and gave similar patterns of disability, psychological status and other symptoms.

**Table 2** Cluster membership at 7 years stratified by the original study cluster (n=155)

| Original study cluster | Number of people in the original study cluster | n (%)* in each cluster (trajectory) at 7 years | | | |
|---|---|---|---|---|---|
| | | No or occasional pain | Persistent mild pain | Fluctuating pain | Persistent severe pain |
| Recovering | 57 | 34 (59) | 18 (32) | 3 (5) | 2 (4) |
| Persistent mild | 51 | 12 (23) | 29 (56) | 8 (15) | 2 (5) |
| Fluctuating | 16 | 1 (7) | 6 (38) | 2 (13) | 7 (42) |
| Severe chronic | 31 | 0 (0) | 4 (12) | 4 (14) | 23 (74) |

*Estimated following multiple imputation.
Weighted κ=0.54 (95% CI 0.42 to 0.65).

**Table 3** Characteristics of cluster membership at 7-year baseline follow-up, groups 1 and 2 (n=155)

| | Cluster (trajectory) from 7-year follow-up analysis | | | | |
| | No or occasional pain | Persistent mild pain | Fluctuating pain | Persistent severe pain | p Value |
|---|---|---|---|---|---|
| In cluster (%) | 31 | 37 | 11 | 21 | |
| Age (years) | 46.3 (43.9 to 48.6) | 47.7 (45.5 to 50.0) | 46.3 (42.1 to 50.6) | 47.0 (43.7 to 50.2) | 0.85 |
| Female | 65% (51 to 80) | 63% (50 to 77) | 68% (43 to 93) | 63% (45 to 81) | 0.99 |
| Pain intensity | 0.8 (0 to 1.8) | 2.3 (1.8 to 2.8) | 4.9 (3.6 to 6.3) | 6.7 (5.8 to 7.6) | <0.001 |
| Leg pain | 42% (26 to 58) | 51% (37 to 65) | 78% (54 to 100) | 83% (68 to 98) | 0.009 |
| Upper body pain | 52% (36 to 68) | 71% (58 to 84) | 88% (71 to 100) | 93% (84 to 100) | 0.004 |
| Disability | 2.0 (0 to 4.1) | 4.3 (3.0 to 5.6) | 8.7 (5.7 to 11.7) | 12.9 (10.5 to 15.3) | <0.001 |
| Anxiety | 5.3 (4.1 to 6.4) | 6.8 (5.6 to 8.0) | 6.5 (4.4 to 8.6) | 8.8 (7.3 to 10.3) | 0.005 |
| Depression | 2.8 (1.8 to 3.8) | 4.9 (3.8 to 6.0) | 4.3 (2.8 to 5.8) | 7.4 (5.9 to 8.8) | <0.001 |
| PHQ-15 | 3.9 (2.6 to 5.3) | 5.0 (3.9 to 6.1) | 7.4 (4.3 to 10.4) | 7.7 (5.8 to 9.7) | 0.006 |
| Insomnia | 27% (12 to 42) | 42% (28 to 57) | 75% (51 to 98) | 80% (65 to 96) | <0.001 |

Figures are mean (95% CI) except for female, leg pain, upper body pain and insomnia, which are percentage (95% CI).
PHQ-15, Patient Health Questionnaire 15.

## DISCUSSION

This study provides unique prospective data on the long-term course of back pain. It suggests that most people remain in a particular pain trajectory, with similar characteristics, when estimated in two periods at the beginning and end of a 7-year period. These findings do not support the hypotheses that there are phases, or degeneration, in the course of back pain over time. Our findings show that widely fluctuating pain is not common (the fluctuating cluster was consistently smallest), and most people have pain patterns varying slightly around their own mean long-term pain. This includes people who recover quickly, and maintain very low (or no) pain, and people who have persistently higher levels of pain. Descriptions of back pain often assume a prevailing pattern of recurrent or fluctuating pain.[23][24] Our findings, and recent qualitative work,[25] provide evidence that these opinions do not give the full picture. However, our study reports pain trajectories among individuals who have sought healthcare, and although recent work identifying general population trajectories of back pain showed trajectories similar to ours,[26] their fluctuating cluster comprised more of the population (35%).

Strengths of the current study include the long-term nature, prospective design, frequent follow-up during study periods, robust analyses and use of validated questionnaire instruments. However, the study did suffer from loss to follow-up, meaning limited number of participants for full analysis. Multiple imputation was used to

**Table 4** Original study baseline characteristics of study participants

| | Full 7-year follow-up (group 1: n=112) | Limited 7-year follow-up (group 2: n=43) | Groups 1 and 2: (n=155) | No 7-year follow-up data available (group 3: n=187) | p Value: groups 1 and 2 vs 3 |
|---|---|---|---|---|---|
| Gender (female)* | 72 (64%) | 28 (65%) | 100 (65%) | 100 (53%) | 0.04 |
| Age (years) | 46.9 (8.3) | 47.0 (7.7) | 47.0 (8.1) | 47.4 (8.2) | 0.63 |
| Pain intensity | 4.4 (2.7) | 4.5 (2.9) | 4.4 (2.8) | 4.7 (2.5) | 0.26 |
| Disability | 9.1 (6.8) | 10.7 (6.8) | 9.5 (6.8) | 10.6 (6.4) | 0.14 |
| CPG IV* | 30 (28%) | 17 (40%) | 47 (31%) | 57 (32%) | 0.86 |
| Anxiety | 8.2 (4.8) | 9.1 (4.6) | 8.5 (4.8) | 8.6 (4.9) | 0.82 |
| Depression | 6.1 (4.4) | 8.4 (4.9) | 6.8 (4.6) | 7.5 (4.8) | 0.15 |
| Duration of pain* | | | | | |
| ≤6 months | 42 (38%) | 14 (33%) | 56 (36%) | 51 (28%) | 0.10 |
| 7–35 months | 23 (21%) | 14 (33%) | 37 (24%) | 48 (26%) | |
| ≥3 years | 46 (41%) | 15 (35%) | 61 (40%) | 85 (46%) | |
| Cluster* | | | | | |
| Recovering | 42 (38%) | 15 (35%) | 57 (37%) | 47 (25%) | 0.10 |
| Persistent mild | 34 (30%) | 17 (40%) | 51 (33%) | 71 (38%) | |
| Fluctuating | 13 (12%) | 3 (7%) | 16 (10%) | 29 (16%) | |
| Severe-chronic | 23 (21%) | 8 (19%) | 31 (20%) | 40 (21%) | |

Figures are mean (SD) except those marked.
*Which are numbers (percentage).
CPG IV, Chronic Pain Grade IV.

investigate the implications of this, and participants included in primary analyses were similar to those excluded, but the possibility of selection bias and residual confounding cannot be ruled out. Although this study had frequent follow-up points during data collection phases, these phases were 7 years apart, and information about trajectories in the interim period is unavailable and therefore unknown.

Few studies have suggested models for long-term change in back pain. Our study gives some support to the model by Raspe et al,[11] as worsening back pain trajectory was significantly associated with more disability, distress, other pains and symptoms, similar to their model of symptom 'amplification'. However, the prospective nature of our study indicates that this 'amplification' is not related to deterioration over time or stages of change, but describes the underlying differences between groups of people whose general pattern of pain does not appear to change over time. In addition, it appears that the spread of pain, further discomforts and depressive symptoms increases fairly consistently with increasing severity of pain trajectory, rather than occurring in discrete stages, as in the amplification model.[11 27] Our results also do not support models of degeneration with age,[12] as clusters do not differ by age. Our findings suggest a new framework model for the long-term course of back pain, comprising four different types of back pain trajectories, each with characteristic pain patterns, disability levels, psychological status and wider symptoms.

New research is emerging on the treatment of back pain according to prognostic risk groups,[9] but questions have been raised about timing of risk group allocation.[28] Our research highlights potentially stable groups of people with different pain trajectories and characteristics. Comparison of the two study phases showed that no cluster changed mean RMDQ score by over 2.5 points (a recommended clinically important change for back pain). This knowledge could improve allocation of treatment according to prognostic risk. However, collecting data over 6 months to allocate treatment is not clinically plausible, and work is needed to identify pain trajectories concisely and accurately. An important implication of our findings is that classifying back pain simply as acute or chronic is insufficient. This is apparent when standard chronic pain definitions would group people with persistent mild symptoms with people who experience constant high levels of pain and other symptoms. Previous work has also highlighted problems defining acute and chronic pain,[25 29] but clinical guidelines are still formulated on this basis.[30 31] Researchers and clinicians should begin to rely less on standard definitions of back pain.

This study raises questions of when, during the life course, trajectory membership is determined. Adolescent trajectories of back pain showed some features similar to the current study (eg, a cluster with very high probability of pain), whereas other trajectories indicated development of a pain condition.[32] Comparable trajectories were also identified for headache, facial pain and stomach pain in the adolescent cohort,[32] which indicates potential applicability of these findings to other conditions, particularly non-specific symptoms.[33 34]

## CONCLUSIONS

We have provided unique evidence on the long-term course of back pain, and suggested a new framework for understanding the course of the condition. There is evidence against phases of change in back pain over time. There are limitations of the study, such as the lack of information about the time between data collection periods, but if the results apply to a significant proportion of back pain patients, there are important clinical implications. First, a large proportion of those who do report initial pain recover quickly, but among those who do not, our results show that many will remain in the same trajectory when assessed several years later. Second, if people in the most severe trajectories could be identified when seeking healthcare, they could be directed to specific targeted treatments. The current study provides substantial new understanding of the long-term course of back pain, and has the potential to have an impact in research and clinical arenas.

**Acknowledgements** The authors would like to thank all the participants and general practitioners who participated in the original and follow-up phases of this study. We thank Professor Peter Croft for his comments on the drafts of the manuscript.

**Contributors** KMD conceived the study and drafted the manuscript. PC and KMD coordinated the data collection. KMD and KPJ analysed the data. KMD had full access to all the data in the study and takes responsibility for the integrity of the data and the accuracy of the data analysis. All authors contributed to the design of the study and revisions, interpreted the data and approved the final version of the manuscript submitted for publication.

**Funding** The work was supported by the Wellcome Trust grant number 083572.

**Competing interests** None.

**Ethics approval** The original study and the 2009–2010 follow-up phase were independently approved by the North Staffordshire Local Research Ethics Committee and South Staffordshire Research Ethics Committee, respectively.

**Provenance and peer review** Not commissioned; externally peer reviewed.

**Data sharing statement** The Arthritis Research UK Primary Care Centre has established data sharing arrangements to support joint publications and other research collaborations. Applications for access to anonymised data from our research databases are reviewed by the Centre's Data Custodian and Academic Proposal (DCAP) Committee, and a decision regarding access to the data is made subject to the NRES ethical approval first provided for the study and to new analysis being proposed. Further information on our data sharing procedures can be found on the Centre's website (http://www.keele.ac.uk/pchs/publications/datasharingresources/) or by emailing the Centre's data manager (primarycare.datasharing@keele.ac.uk).

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
