## [Reviewer comments · BMJ Open]

Some articles will have been accepted based in part or entirely on reviews undertaken for other BMJ Group journals. These will be reproduced where possible.

ARTICLE DETAILS

TITLE (PROVISIONAL)	Long-term trajectories of back pain: cohort study with seven year follow-up
AUTHORS	Dunn, Kate; Campbell, Paul; Jordan, Kelvin

VERSION 1 - REVIEW

REVIEWER	Axén, Iben Karolinska Institute, IMM Institute of Environmental Medicine
REVIEW RETURNED	24-Sep-2013

GENERAL COMMENTS	I lack some information regarding the resulting clusters. (Could possibly be found in the Appendix, which I cannot find.) For instance, the first cluster is mainly no pain with occasional mild episodes. What is the definition of occasional? What is the definition of episode? As there is an ongoing scientific discussion regarding episode, this may be important to define. Likewise, for the fluctuating cluster, maybe some information regarding the duration of fluctuation would add some descriptive value. This could also add some explanatory value to the size of this cluster when comparing to other studies.
--

REVIEWER	Werner, Erik Research Unit for General Practice, Uni Health, Bergen, Norway
REVIEW RETURNED	02-Oct-2013

GENERAL COMMENTS	This is an interesting paper on long term observation of LBP patients. My major concern regarding this paper is in my opinion an exaggeration of the findings. The results of this study is that people having minor back pain at one point also have minor pain seven years later, and that those with severe pain also report severe pain seven years later. Of the period inbetween we do not know anything. The authors assume from these two data collection points that the pain has followed the same pattern all the way, which is somewhat as opposed to previous research. To make such interpretation of the study one would wish to have several points of data collection across the time. What about the frequency, duration and intensity of the recurrent back pain during the seven years? In fact, only 45% of the original 342 participants are followed in the study, which is an important limitation. Of minor comments, the description of the four trajectories is not identical in the abstract and the manuscript, and the explanation of these trajectories is not clarified until the result section. One would wish this explanation already from the start. On p 13 line 13 underlying differences between the groups are
--

	suggested as reasons for belonging to one or the other group. This statement should be elaborated. On p 12 line 29 there is an 'are' too much. The English language is certainly good, but the paper would benefit from editing the presentation of the results. The statistics seem impressive and good to me, but with limited expertise in this field and some advanced statistics in the study, I feel uncomfortable by reviewing this part.
--	--

VERSION 1 – AUTHOR RESPONSE

Reviewer Name Iben Axén

I lack some information regarding the resulting clusters.

- We have added more detail to the paper about the resulting clusters.

(Could possibly be found in the Appendix, which I cannot find.)

- Apologies, the appendix has now been included.

For instance, the first cluster is mainly no pain with occasional mild episodes. What is the definition of occasional? What is the definition of episode? As there is an ongoing scientific discussion regarding episode, this may be important to define. Likewise, for the fluctuating cluster, maybe some information regarding the duration of fluctuation would add some descriptive value. This could also add some explanatory value to the size of this cluster when comparing to other studies.

- As the identification of the clusters was based on the statistical analyses, there were no a-priori definitions of terms such as occasional, episodes etc. However, the estimated probabilities of pain provided in Table 1 and summarised on page 10 give the required information. These monthly probabilities of pain can be interpreted to describe the occurrence of pain, for example, a probability of mild-moderate pain of 0.13 at baseline for the first Cluster indicates that one in every 8 people in that group are likely to have experienced mild-moderate pain that month. We have added further information about this to the results to aid interpretation of the data.

Reviewer Name Erik L. Werner

This is an interesting paper on long term observation of LBP patients. My major concern regarding this paper is in my opinion an exaggeration of the findings. The results of this study is that people having minor back pain at one point also have minor pain seven years later, and that those with severe pain also report severe pain seven years later. Of the period in-between we do not know anything. The authors assume from these two data collection points that the pain has followed the same pattern all the way, which is somewhat as opposed to previous research. To make such interpretation of the study one would wish to have several points of data collection across the time. What about the frequency, duration and intensity of the recurrent back pain during the seven years?

- It is true that we cannot report information about the pain experience during the time between the two study periods, and we have clearly pointed out this limitation in the article summary (page 3), as well as in the discussion (page 13). We have also added information on this limitation to the abstract (page 2). However, we would like to reiterate that we do not simply have two data collection points, rather we have two periods of intensive data collection, each spanning 6 months, which we would argue adds substantially to the validity of the findings and our interpretation. We also maintain that this is not opposed to previous research, as there has been no previous studies looking at trajectories of back pain using data from two (or more) periods of data collection. The only other studies have identified trajectories from one study period, or have used single data collection points, which could

lead to important misclassification (e.g. someone reporting mild-moderate pain at a single time point could be classified into any of the 4 trajectories) – only repeated measurements leading to identification of trajectories can increase the likelihood of correct classification.

In fact, only 45% of the original 342 participants are followed in the study, which is an important limitation.

- We have acknowledged this limitation in the article summary (page 3), the discussion (page 13) and the abstract (page 2).

Of minor comments, the description of the four trajectories is not identical in the abstract and the manuscript, and the explanation of these trajectories is not clarified until the result section. One would wish this explanation already from the start.

- Apologies for the inconsistent descriptors of the trajectories in the abstract and manuscript. This has now been corrected. We are unable to describe (explain) the trajectories until the results section, as they are identified as a result of the statistical analyses.

On p 13 line 13 underlying differences between the groups are suggested as reasons for belonging to one or the other group. This statement should be elaborated.

- This has now been amended (discussion, page 15, 1st paragraph), and more detail has been added to the results (page 12).

On p 12 line 29 there is an 'are' too much.

- Thank-you. We have removed the additional 'are'.

The English language is certainly good, but the paper would benefit from editing the presentation of the results.

- We have now amended our description of the results, and hope that this improves the presentation.

The statistics seem impressive and good to me, but with limited expertise in this field and some advanced statistics in the study, I feel uncomfortable by reviewing this part.

- An appendix has now been included that includes further information about the statistics for the reader.

Although I recommend the paper accepted with minor revision, I do think my concerns are of importance. A modification of the interpretation of the study will however not be very time consuming to carry out.

- Thank you for your suggestions, we hope that we have modified the manuscript sufficiently.

VERSION 2 – REVIEW

REVIEWER	Werner, Erik Research Unit for General Practice, Uni Health, Bergen, Norway
REVIEW RETURNED	28-Oct-2013

GENERAL COMMENTS	There has only been very small changes on the paper from the first review and my concerns regarding the interpretation of the results stand. In my opinion the results show the severity of back pain at two points, which seem to follow a pattern, but the period in between these points is left unknown. This limitation is not sufficiently recognised in the conclusion.
--

VERSION 2 – AUTHOR RESPONSE

We have now amended the manuscript to make it even clearer that the data collection periods were several years apart, and that we do not know what happened to people in the interim period. This is now stated in the abstract conclusions, in the strengths and limitations of the article summary, in the first and 2nd paragraphs of the discussion, and in the final conclusions. We hope that this addresses the concerns of the reviewer.